# Chitin Nerve Conduits with Three-Dimensional Spheroids of Mesenchymal Stem Cells from SD Rats Promote Peripheral Nerve Regeneration

**DOI:** 10.3390/polym13223957

**Published:** 2021-11-16

**Authors:** Ci Li, Meng Zhang, Song-Yang Liu, Feng-Shi Zhang, Teng Wan, Zhen-Tao Ding, Pei-Xun Zhang

**Affiliations:** 1Department of Orthopedics and Trauma, Peking University People’s Hospital, Beijing 100044, China; drlici@bjmu.edu.cn (C.L.); mengzh2008@bjmu.edu.cn (M.Z.); 1911110343@pku.edu.cn (S.-Y.L.); xmx066@pku.edu.cn (F.-S.Z.); tengwan.med@hotmail.com (T.W.); zhentao_ding@stu.pku.edu.cn (Z.-T.D.); 2Key Laboratory of Trauma and Neural Regeneration, Peking University, Beijing 100044, China; 3National Center for Trauma Medicine, Peking University People’s Hospital, Beijing 100044, China

**Keywords:** peripheral nerve injury, chitin, peripheral nerve conduits, bone marrow mesenchymal stem cells, spheroid

## Abstract

Peripheral nerve injury (PNI) is an unresolved medical problem with limited therapeutic effects. Epineurium neurorrhaphy is an important method for treating PNI in clinical application, but it is accompanied by inevitable complications such as the misconnection of nerve fibers and neuroma formation. Conduits small gap tubulization has been proved to be an effective suture method to replace the epineurium neurorrhaphy. In this study, we demonstrated a method for constructing peripheral nerve conduits based on the principle of chitosan acetylation. In addition, the micromorphology, mechanical properties and biocompatibility of the chitin nerve conduits formed by chitosan acetylation were further tested. The results showed chitin was a high-quality biological material for constructing nerve conduits. Previous reports have demonstrated that mesenchymal stem cells culture as spheroids can improve the therapeutic potential. In the present study, we used a hanging drop protocol to prepare bone marrow mesenchymal stem cell (BMSCs) spheroids. Meanwhile, spherical stem cells could express higher stemness-related genes. In the PNI rat model with small gap tubulization, BMSCs spheres exhibited a higher ability to improve sciatic nerve regeneration than BMSCs suspension. Chitin nerve conduits with BMSCs spheroids provide a promising therapy option for peripheral nerve regeneration.

## 1. Introduction

Peripheral nerve injury (PNI) is a worldwide clinical disease that causes sensory and motor deficits, which may affect the health and life quality of patients [1]. A complete disruption of the nerve fiber is the most severe form of PNI. Reconstruction of the anatomical structure of peripheral nerve tissue is the primary condition for functional recovery. Presently, most nerve transection injuries are treated by epineurium neurorrhaphy. However, microscopic epineurium neurorrhaphy may lead to inaccurate connection of nerve fibers and neuroma formation, which may be unable to effectively restore peripheral nerve function [2]. Our team found that small gap tubulization can overcome the disadvantages of traditional epineurial neurorrhaphy and provide a suitable treatment method for PNI [3].

Primary repair with tubulization suture is a promising management for nerve discontinuities. The artificial peripheral nerve conduits have been developed to bridge the two stumps of the defected nerves [3]. To facilitate the anastomosis of the nerve stumps and the conduit and reduce the occurrence of postoperative complications, the peripheral nerve conduits should have several characteristics, such as high flexibility and malleability, collapse resistance, low tissue adhesion, and sufficient transparency [4,5]. Recently, tissue engineering has focused on various natural compounds with biological effects. Chitin is one of the natural compounds with a wide range of sources and unique biological characteristics. Several studies have reported that chitin and chitosan can potentially interact with nerve regeneration-related cells and the neural microenvironment, thereby improving axon regeneration and reducing neuroma formation [6,7,8]. According to different applications, chitin and chitosan can be made into hydrogels, films, microspheres and fibers, etc., reflecting chitin’s great potential as a matrix material for tissue engineering [9,10]. However, the dissolution conditions of chitin are harsher than chitosan, and the films based on chitosan are usually very fragile [11,12]. These factors limit the fabrication of chitosan peripheral nerve conduits. On this basis, we solidified the chitosan solution with NaOH solution, and then made the chitin peripheral nerve conduits after performing the acetylation procedure.

Recently, more and more evidence has indicated that introducing three-dimensional (3D) culturing overcomes some limitations of mesenchymal stem cells (MSCs) suspensions transplantation. A variety of beneficial characteristics of MSCs can be maintained in 3D culture because the interactions of cell-to-cell and cell-to-matrix can be maintained better in a 3D environment [13]. Among cell scaffolds, gels and spheroid fusion, the 3D spheroids have attracted widespread attention in terms of uniform cell distribution, high fusion potential, and closeness to the natural cell microenvironment [14,15,16].

In the present study, we formed the Bone marrow mesenchymal stem cells (BMSCs) spheroids by hanging drop and examined their stemness potential. In addition, chitin conduit micromorphology, mechanical properties, and biocompatibility were characterized in vitro. Then, in the conduit small gap tubulization model, BMSCs spheroids were administered in the gap between chitin conduits and nerve stumps. The curative effects of BMSCs spheroids were evaluated by motor function, conduction function, and morphology of regenerated nerves.

## 2. Materials and Methods

### 2.1. Isolation and Culture of BMSCs

The isolation and culture of BMSCs were performed as previously described [17]. Briefly, after euthanasia and sterilization, the femurs of SD rats within 28 days of birth were collected under sterile conditions. Then, a 10 mL syringe fitted with a 25-gauge needle was used for flushing the bone marrow from the femurs. Then, the cells were cultured in Dulbecco’s Modified Eagle Medium: F-12 (DMEM/F-12, Thermo Fisher Scientific, Waltham, MA, USA) with 10% fetal bovine serum (FBS, Thermo Fisher Scientific) and 1% penicillin-streptomycin (Solarbio, Beijing, China). After 24 h, the medium was changed to remove non-adherent cells. BMSCs were cultured in a humidified 37 °C cell incubator with 5% CO_2_. The BMSCs firstly formed colonies and were considered as passage 0 when typical vortex arrangement appeared. Passages 2 to 6 were used in the following experiments.

### 2.2. Identification of BMSCs

The identification of BMSCs was proved by their differentiation ability and surface markers [18]. Second-passage MSCs were seeded in two 6-well plates at a density of 2.5 × 10^5^ cells per well for osteogenesis and lipid differentiation, respectively. Briefly, we applied osteogenesis induction medium (Cyagen Biosciences Inc., Sunnyvale, CA, USA) and adipogenic induction medium (Cyagen Biosciences Inc., Sunnyvale, CA, USA) to culture BMSCs for 21–28 days. After induction, BMSCs in each well were washed with PBS and fixed with 4% paraformaldehyde for 15–30 min at room temperature. Then, alizarin red S (MilliporeSigma, Burlington, MA, USA) and Oil Red O (MilliporeSigma) were used to stain calcium nodules and lipid droplets according to the instructions. All differentiated cells were observed by optical microscope (Leica, Wetzlar, Germany).

At passage 2, BMSCs were digested with trypsin and washed with PBS three times. Subsequently, anti-CD29 (0.2 µg/10^6^ cells, Biolegend, San Diego, CA, USA), anti-CD34 (1:50, Bioss, Beijing, China), anti-CD45 (0.2 µg/10^6^ cells Proteintech, Chicago, IL, USA), and anti-CD90 (0.2 µg/10^6^ cells, Thermo Fisher Scientific) were used to incubate BMCSs suspension for 1 h. Following centrifugation at 1500 rpm for 5 min, the supernatant was discarded. Then, flow cytometry (CytoFLEX, Beckman, Brea, CA, USA) detected and analyzed the characteristics of BMSCs.

### 2.3. Spheroids Formation

BMSCs spheroids were generated by applying the hanging-drop culture technique [19]. Briefly, a suspension of the single BMSCs was prepared as a typical procedure. A total 20 μL of cell suspension containing about 2 × 10^4^ cells was placed evenly without touching each other on the inside of the lid of the sterile petri dish. After that, the lid was quickly placed on the petri dish containing 5–10 mL sterile PBS. About 24 h later, BMSCs spheroids were formed in cell suspension drops, and the BMSCs spheroids continued to be cultured in the hanging drops.

### 2.4. Morphological Observation of Cell Spheroids

After the cell spheroids were grown in the hanging drops for 72 h, the spheroids were collected and washed washed twice by PBS. Then, the cell spheroids were fixed with 4% paraformaldehyde (MilliporeSigma) for 15 min and permeabilized with 0.5% Triton X-100 (MilliporeSigma) for 10 min. Phalloidin (Solarbio) was used to label the cytoskeleton. In order to make the reagents fully penetrate into the spheroids, phalloidin (1:200, MilliporeSigma) and DAPI (1:1000, MilliporeSigma) were stained with the cell spheroids overnight. After being washed three times by PBS, immunofluorescence images were obtained by a laser scanning confocal microscope.

### 2.5. Quantitative Real-Time Polymerase Chain Reaction (qRT-PCR)

The qRT-PCR was performed to evaluate the stemness-related gene expression of BMSCs sheets and BMSCs spheroids [20]. Briefly, the total RNA was extracted with the equal cell number of BMSCs sheets and BMSCs spheroids with miRNA kit (Tiangen, Beijing, China). Then, complementary DNA was synthesized by reverse transcription. The primers of Nanog, Sox2, and POU5F1 and internal reference primers for glyceraldehyde 3-phosphate dehydrogenase (GAPDH) were designed as follows: Nanog, Sox2, POU5F1, and GAPDH. Nanog: forward primer (5′-3′) is CCGTGTTGGCTGCATTTGTCTG, reverse primer (5′-3′) is TGGAGTAGGGTGGGTGTGTGAG; Sox2: forward primer (5′-3′) is CCAGCTCGCAGACCTACATGAA, reverse primer (5′-3′) is GCCTCGGACTTGACCACAGA; POU5F1: forward primer (5′-3′) is CACCACACTCTACTCGGTCCCT, forward primer (5′-3′) is CTTGCCTTGGCTCACCTCATCC; GAPDH: forward primer (5′-3′) is GCCATCACTGCCACTCAGAAGA, reverse primer (5′-3′) is ATGACCTTGCCCACAGCCTTG. Cycling conditions were: 95 °C for 1 min; followed by 40 cycles of 95 °C for 5 s, 60 °C for 15 s, and finally 75 °C for 45 s.

### 2.6. Fabrication of the Chitin Nerve Conduits

The construction process of the chitin nerve conduits has been described previously [21]. Briefly, 4% (*w*/*v*) chitosan solution was prepared by dissolving chitosan powder in 2% glacial acetic acid. Then, a glass rod stirred the chitosan solution until the solution was uniform. An ultrasonic cleaner (Jinli, Shanghai, China) was used to remove air bubbles. A specific mold with a diameter of 1.5 mm was slowly immersed in the chitosan solution. The 5% sodium hydroxide solution was used to solidify the chitosan substrate when the chitosan solution covered the surface of the mold uniformly. After the solidified substrate was dried at room temperature (RT), the substrate was acetylated with acetic anhydride for 30 min. After that, the chitin conduits were carefully peeled off the mold. The hollow conduits were kept in 75% ethyl alcohol for further use (Figure 1).

### 2.7. Characterization of Chitin Nerve Conduits

#### 2.7.1. Scanning Electron Microscopy

As described previously, a scanning electron microscope (JEOL, Tokyo, Japan) was used to observe the surface morphology of the chitin conduits [22]. Briefly, thin films were extracted from the chitin conduits and freeze-dried overnight. Then, the chitin films were fixed on the specimen platforms and coated with gold. The photographs of the chitin films were obtained at a magnification of 10,000× *g*.

#### 2.7.2. Water Contact Angle Measurement

A water contact angle measurement platform (Dataphysics, Filderstadt, Germany) was used to measure the water contact angle of chitin conduits [23]. Firstly, the chitin film was tiled on the specimen conduit. Then, 2 μL ultrapure water was dropped on the films. The process was recorded by a high-speed camera (Dataphysics). Three specimens were analyzed with three measurements for each surface.

#### 2.7.3. Fourier Transform Infrared Spectroscopy

The films were peeled from the surface of the chitin conduits carefully. As described previously, infrared spectra of the films were obtained by using a Fourier-transform infrared spectrometer (Thermo Fisher Scientific) [24]. Briefly, the dried chitin and chitosan films were placed on the device and scanned by the equipment. The wavelength range was from 600 to 4000 cm^−1^ and the spectral resolution was 4 cm^−1^.

#### 2.7.4. Tensile Stress Test

As described previously, the tensile stress of chitin conduits was evaluated by the tensile tester (Mark-10, New York, NY, USA) [25]. Briefly, both ends of each conduit were fixed to clamps. The machine tightened the conduit at a speed of 13 mm/min until the conduit was disconnected. All the examinations were performed in the same external conditions (RT = 25 °C, Air humidity = 50%).

#### 2.7.5. Live/Dead Staining

The biocompatibility of the chitin surface was investigated by live/dead cell staining [26]. Briefly, Schwann cells (SCs, isolated from the sciatic nerves of rats) were seeded on the chitin film surface at a density of 1.0 × 10^5^ cells/cm^2^. Then, 72 h after implantation, the SCs were washed by PBS three times. According to the instructions of the live/dead cell staining kit (Solarbio), the cells were stained with Calcein-AM (a green fluorescent dye for live cells, Ex/Em = 488/515 nm) and propidium iodide (PI, a red fluorescent dye, Ex/Em = 535/617 nm). After incubation for 15 min at 37 °C, the cells samples were washed by PBS three times gently. The cells samples were then observed in a fluorescence microscope (Leica).

### 2.8. Rat Model of Sciatic Nerve Injury and Cell Transplantation

All animal experiments were approved by the Institutional Animal Care and Use Committee of the Peking University People’s Hospital (Approval No. 2020PHE098), and care of the animals was in accordance with the Laboratory Animal Guideline for Ethical Review of Animal Welfare of China (GB/T 35892-2018). Fifteen adult Sprague Dawley (SD) rats (weighing 200–220 g and aged 6–8 weeks) were obtained from Beijing Vital River Laboratory Animal Technology.

All the animals were divided into 3 groups (*n* = 5/group): the hollow chitin conduits (Control group), chitin conduit-integrated BMSCs suspension (BMSCs group), and chitin conduit-integrated BMSCs spheroids group (BMSCs spheroids group). As described previously [21], the skin of the right hind of the rat was cleaned and disinfected before surgery. The rats in each group were anesthetized with 3% sodium pentobarbital (30 mg/kg) by intraperitoneal injection. After that, the right sciatic nerve was exposed and transected at 7 mm above the sciatic nerve fork. The nerve stumps were bridged and a 2 mm gap with chitin nerve conduit under a surgical microscope (Leica) was maintained using 10-0 nylon sutures. As shown in Figure 2, after implanting the chitin nerve conduit, rats in the Control group were injected with 20 μL PBS, rats in the BMSCs group were injected with the BMSCs suspensions (5 × 10^5^ cells in 20 μL PBS), and rats in the BMSCs spheroids groups were injected with 25 spheroids in 20 μL PBS. Finally, the muscles and skin were sutured carefully. After waking, all the animals could eat and drink freely in the cages.

Walking track analysis was performed 8 weeks after surgery [27]. The CatWalk XT 10.6 gait analysis system (Noldus, Wageningen, The Netherlands) was used to record and analyze the footprints of the rats. In brief, all the rats were already familiar with the testing environment before the experiment. Then, the rats in each group passed a closed runway composed of a bottom glass plate and clapboards. The footprint of each rat in each group was collected by a high-speed camera. The Sciatic Function Index (SFI) in each animal was calculated by the following formula: SFI = −38.3 × (EPL − NPL)/NPL + 109.5 × (ETS − NTS)/NTS + 13.3 × (EIT − NIT)/NIT − 8.8. EPL represents the experimental paw length and NPL means the normal paw length (the distance from the heel to the top of the third toe). ETS is the experimental toe spread and NTS is the normal toe spread (the distance from the first to the fifth toe). EIT means the experimental intermediary toe spread and NIT means the normal toe spread (the distance from the second to the fourth toe). In general, the SFI oscillates around 0 for normal nerve function, whereas around −100 SFI represents total dysfunction.

### 2.9. Electrophysiological Examination

Rats were anesthetized by an intraperitoneal injection of 3% sodium pentobarbital (30 mg/kg) [28]. An electrophysiological instrument (Oxford Instruments, Oxford, UK) was performed to evaluate the compound muscle action potential (CMAP). The stimulating electrode was placed 5 mm from the proximal end of the nerve conduits. The recording electrodes were attached to the gastrocnemius muscle. Then, the CMAP amplitudes and latencies were evoked by a rectangular pulse (stimulus intensity 0.9 mA, pulse duration 0.1 ms) and recorded, separately.

### 2.10. Neurohistological Analysis 

The regenerated sciatic nerves were harvested from each rat at 8 weeks after operation [21]. Then, the samples were fixed in 2.5% glutaraldehyde at 4 °C for 2 h. The samples were stained with 1% samarium acid, dehydrated in an acetone gradient, embedded in epoxy resin, and cut into 70 nm ultra-thin sections. After that, sections were stained with uranyl acetate and lead citrate and observed by transmission electron microscopy (Philips, Amsterdam, The Netherlands). The diameter of the regenerated axon and the thickness of the remyelin sheath were measured using ImageJ software (National Institutes of Health, Bethesda, MD, USA).

### 2.11. Statistical Analysis

Statistical data analyses were conducted using GraphPad Prism software (version 7.04; San Diego, CA, USA). Student’s t test was used to compare the two groups. Before analysis, the data of multiple groups were tested for normal distribution by Kolmogorov–Smirnov and Shapiro–Wilk tests. The data that conformed to normal distribution were presented as mean ± Standard Error Estimate (SEM). If not, the data were presented as median with interquartile range. One-way analysis of variance (ANOVA) was employed for analysis when the data conformed to normal distribution. If not, a nonparametric test was used to analyze the data. *p* < 0.05 was defined as a significant difference.

## 3. Results

### 3.1. Phenotype Identification of BMSCs

As shown in Figure 3a, adhered cells exhibited a vortex distribution and shuttle-shaped morphology. After 21 days of osteogenic differentiation, the alizarin red S staining assay demonstrated the osteogenesis of BMSCs. As shown in Figure 3b, there were accumulation of calcium nodules formed by differentiated cells. After 28 days of adipogenic differentiation, Oil Red O staining showed apparent lipid droplets within the cells (Figure 3c). The flow cytometry showed the expressions of cell surface antigens (Figure 3d). The results showed positive-antigen CD29 and CD90 were high expression (both >95%) and negative-antigen CD34 and CD45 were low expression (both <5%). In conclusion, high purity MSCs were extracted and further used in experiments.

### 3.2. BMSCs Spheroids Exhibited Enhanced Stemness Related Potential

As shown in Figure 4a, BMSCs spheroids were formed in hanging drops and spheroids cultured in medium lost their typical BMSC shape. From Figure 4b, the average diameter of MSCs spheres measured was about 190.4 ± 9.495 μm (*n* = 3 spheres). Confocal microscopy imaging showed the morphology of BMSCs spheres (Figure 4c). The green in the sphere represents the cytoskeleton labeled with phalloidin, and the blue was the nucleus. Meanwhile, the BMSCs sphere was not a standard sphere, but the blue–green crosslink spheroid could reflect the close connection between cells.

For stem cell-based therapy, the stemness potential of the implanted cells is a vital property for their therapeutic benefit. Thus, the stemness capability of BMSCs spheroids was investigated by qRT-PCR. After 72 h of suspension culture, total RNA was extracted from normal cell culture and stable BMSCs spheres. Compared with ordinary cell culture, significantly higher expression of stemness-related genes (Nanog, Sox2, and POU5F1) was observed in the cell spheres (Figure 4e–g).

### 3.3. Characteristics of Chitin Conduits

Chitin nerve conduits of different lengths can be obtained as required. As shown in Figure 5a, the length of the conduits used in this study was 6 mm, and the inner diameter was 1.5 mm. The conduits were immersed in liquid for storage, showing a transparent state. Meanwhile, the nerve conduit was soft and flexible (Figure 5b). As shown in Figure 5c, The scanning electron microscope image showed that the surface of chitin material was rough. There was no regular pore structure formed on the inner surface of the chitin conduit.

As shown in Figure 6a–c, the chitin nerve conduits had an initial length of 10 mm and a stretched length of 16.8 mm. According to the force–displacement curve (Figure 6d), the chitin nerve conduits presented a suitable tensile property for the connection of peripheral nerve stumps.

As shown in Figure 7, a broad band was observed at 3000–3500 cm^−1^ due to the stretching vibration of the –OH group. The peak at 2922 cm^−1^ might be the characteristic stretching vibrations for C-H groups of organic compounds. Chitin exhibited characteristic bands of carbonyl (C=O) and amine group (–NH_2_) at 1637 cm^−1^ and 1549 cm^−1^, which were not completely consistent with chitosan. The C-O-C stretching might cause the band to appear at 1035 cm^−1^.

The hydrophilicity of the chitin conduit surface was reflected by measuring the contact angle formed by the water droplet with the conduit surface. As shown in Figure 8, the water contact angle of the original chitin surface was about 35 ± 5.13°.

### 3.4. Live/dead Staining

As shown in Figure 9, live/dead staining revealed that the chitin surface possessed good biocompatibility. The live SCs were labeled green by Calcein-AM, while dead cells appeared red after PI staining. After culturing for 3 days, the number of dead cells in the field of view was much smaller than that of live cells.

### 3.5. Effect on the Recovery of Hindlimb Motor Function

SFI is an objective indicator to detect the recovery of motor function in rats. During the test, the paw prints of the rats running through the path were recorded, and the SFI results of the rats were calculated according to the formula. The representative images of footprints were shown in Figure 10a, in which the morphology of the right footprint in the BMSCs spheroids group was similar to that in the left footprint. As shown in Figure 10b, the SFI results in the BMSCs and BMSCs spheroids groups were significantly lower than in the control group at 4 weeks after surgery (*p* < 0.05). The SFI between the BMSCs and BMSCs sphere groups did not show a significant difference. At 8 weeks after the operation, SFIs in the BMSCs spheroids group were significantly lower than the other groups (*p* < 0.05).

### 3.6. Electrophysiological Parameters of Injured Sciatic Nerves

The electrophysiological examination was performed 8 weeks after surgery. Figure 11a shows the representative latency and the amplitude of CMAP in all the groups. As shown in Figure 11b,c, 8 weeks after surgery, CMAP latency was obviously lower in the BMSCs spheroids group than in the other groups (*p* < 0.05). CMAP latency was lower in the BMSCs group than in rats without treatment (*p* < 0.05). CMAP amplitude was significantly higher in the BMSCs and BMSCs spheroids groups than in the Control group (*p* < 0.05). The CMAP amplitude of the BMSCs spheroids group was significantly higher than that of the BMSCs group.

### 3.7. Histological Evaluation of Regenerative Nerves

TEM was used for the visualization of the axonal morphologies at the distal end (Figure 12a). According to the TEM images, the thickness of the myelin sheath in the BMSCs spheroids group was larger than the other groups (*p* < 0.05). Moreover, the mean diameter of remyelinated nerve fibers was significantly larger in the BMSCs and BMSCs spheroids groups than in the Control group (*p* < 0.05). At the same time, there was no apparent difference between the two BMSCs treatment groups (Figure 12b,c).

## 4. Discussion

PNI can lead to sensory and motor dysfunction, complex and refractory complications, and even paralysis, which causes a reduction in the quality of life of patients [29]. Epineurial neurorrhaphy is a common method in the clinical treatment of PNI. Nerve fiber misconnection and neuroma formation are common complications of epineurial neurorrhaphy. Even if the peripheral nerves are sutured in time, it is still difficult to obtain satisfactory functional recovery [30]. Employing nerve conduits to maintain small gaps when bridging nerve stumps not only improves the accuracy of nerve fibers docking but also protects the microenvironment stably [31]. Previous studies have demonstrated that chitin is a biomaterial with favorable biocompatible and flexible properties to fabricate nerve conduits [32,33].

Chitin, one of the most abundant natural amino polysaccharides, is mainly derived from crustaceans in the ocean and is an important component of insect exoskeletons and fungal cell walls [12]. Chitin is a semi-crystalline homopolymer of β-(1-4)-linked N-acetyl-D-glucosamine. Generally, there are three different polymorphs of chitin in structure, which are the α-chitin, the β-chitin, and the γ-chitin [34]. Chitin can be degraded in vivo and has high affinity with multiple cells. Thus, chitin has been widely used as a vital biomaterial in the field of tissue engineering [35]. In this study, chitosan was used as the raw material for the nerve conduits. Under the action of sodium hydroxide, the texture of the chitosan solution on the mold becomes hard so that it can be fixed on the mold. The subsequent acetylation of acetic anhydride once again increases flexibility. A partially acetylated peripheral nerve conduit was constructed. This method successfully provides a simple way for producing chitin nerve conduits. Meanwhile, some of the chemical reagents used in the construction process can be eliminated by later alcohol soaking. 

Here, we investigated the characteristics of chitin conduits. Figure 3a shows that we successfully synthesized a chitin conduit that can be trimmed as demanded. The scanning electron microscope image showed the rough surface of chitin (Figure 3b). The results of tensile stress show that chitin conduits are suitable for nerve tissue repair because the elastic modulus of chitin nerve conduits is greater than that of nerve tissue [36]. Consistent with a previous study, the spectrum of chitin films was not utterly similar to characteristic peaks of chitosan [24]. The contact angle on the surface of the chitin film is an acute angle. In addition, the results of live/dead staining of SCs and the morphology of the cells on the chitin membrane show that the chitin nerve conduit has good biocompatibility.

The regeneration of injured peripheral nerves is a dynamic process requiring various cell–cell responses and cytokines to stimulate axon growth, intracellular remodeling, and functional recovery [37,38,39,40]. Therefore, a single chitin application is unlikely to meet the requirements of PNI patients adequately. Nowadays, stem cell-based tissue engineering techniques have achieved remarkable progress in the field of peripheral nerve regeneration due to stem cells participating in the process of tissue formation [41]. Stem cells-based transplantation therapies involve delivering dispersed or clumped stem cells with or without biological carriers to promote tissue repair. Scattered stem cells directly implanted into the injured area have the phenomenon that the survival rate is reduced and the apoptosis rate is increased. Moreover, fixing the stem cell suspension in the designated area is difficult, and these unfavorable factors limit the therapeutic effects [42]. Studies have shown that stem cell spheres can better fit the modes of cell–cell communication and cell–extracellular matrix interaction in vivo [43,44]. Compared with stem cells in flat culture, spherical stem cells show significant advantages in regeneration. MSCs spheroids are more capable of adapting to a hypoxic environment. A variety of anti-inflammatory genes, which are beneficial to regulate the activation state of macrophages during the regeneration process, can be expressed by MSCs spheroids [44,45]. In the present study, the BMSCs were assembled into spheroids by hanging drop. In the fluorescence image of the cytoskeleton labeled with phalloidin, the BMSCs spheroids formed tight outer shells. It is known that a critical characteristic of MSCs is their multilineage differentiation potentials. Consistent with previous studies [46], the expression of pluripotency marker genes (NANOG, Sox2, and POU5F1) in BMSCs spheroids was higher than that in BMSCs sheets. In the chitin nerve conduits and the peripheral nerve stumps, the space is so small that other fillers such as hydrogel will hinder nerve regeneration. We speculate that BMSC spheroids may survive better and express higher stemness than dispersed BMSCs in the hypoxia and nerve regeneration-inducing environment, thus achieving the therapeutic effects.

In vivo, the recovery of SFI in the BMSCs spheroids group was superior to the control and BMSCs groups. This result could be attributed to the local application of BMSCs spheroids in the small gap after PNI. Electrophysiological measurements were used to evaluate the nerve conduction function. The lower CMAP latency and higher CMAP amplitude in the BMSCs spheroids group showed superior results, indicating that the BMSCs spheroids actively supports the nerve fibers to complete the reinnervation of the target muscles. Histological evaluation of TEM further intuitively confirmed the results. We found that BMSCs spheroids could accelerate nerve regeneration and promote the re-myelin process in the 2 mm microenvironment.

Previous studies showed that chitosan membranes could be used to format stem cell spheroids and improve their properties of migration and self-renewal [47]. However, the biological effect of the chitin peripheral nerve conduit on the BMSCs sphere is not addressed. The mechanisms where the BMSCs spheroids enhance stemness properties and other features such as angiogenesis, migration, and directed differentiation in PNI will be studied deeply in future research.

## 5. Conclusions

In summary, the chitin nerve conduits prepared by acetylation of chitosan has good biocompatibility. Chitin nerve conduits can bridge nerve stumps and have potential application in nerve tissue engineering. In addition, the superior differentiation capacity is an attractive feature of the BMSCs spheroids. The nerve conduit small gap tubulization cooperating with BMSCs spheroids transplantation can be considered as a possible application in promoting peripheral nerve regeneration. Such a novel approach may have great clinical potential for treating PNI.

## Figures and Tables

**Figure 1 polymers-13-03957-f001:**
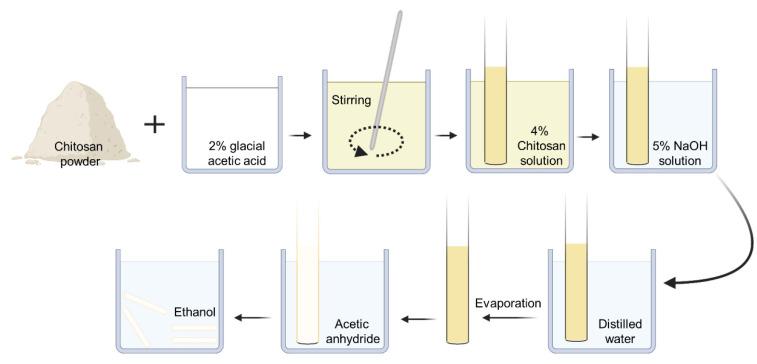
The construction process of chitin nerve conduits.

**Figure 2 polymers-13-03957-f002:**
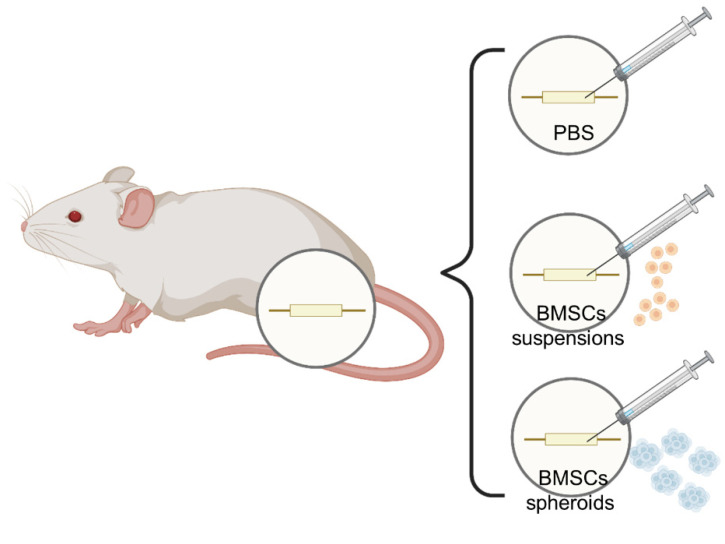
Schematic diagram of each group of nerve conduits and implants.

**Figure 3 polymers-13-03957-f003:**
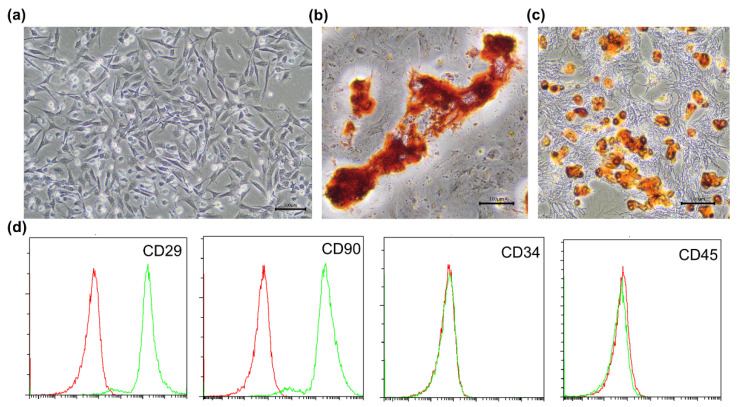
Identification of BMSCs. (**a**) Typical vortex distribution of BMSCs. (**b**) Alizarin red S staining revealed that osteogenic capacity of BMSCs. (**c**) Oil Red O staining showed adipogenic differentiation of BMSCs. (**d**) Expression of BMSCs surface markers CD29, CD34, CD45, and CD90 detected by flow cytometry. Red lines indicated the blank control BMSCs, and green lines indicated the antibodies-bound BMSCs.

**Figure 4 polymers-13-03957-f004:**
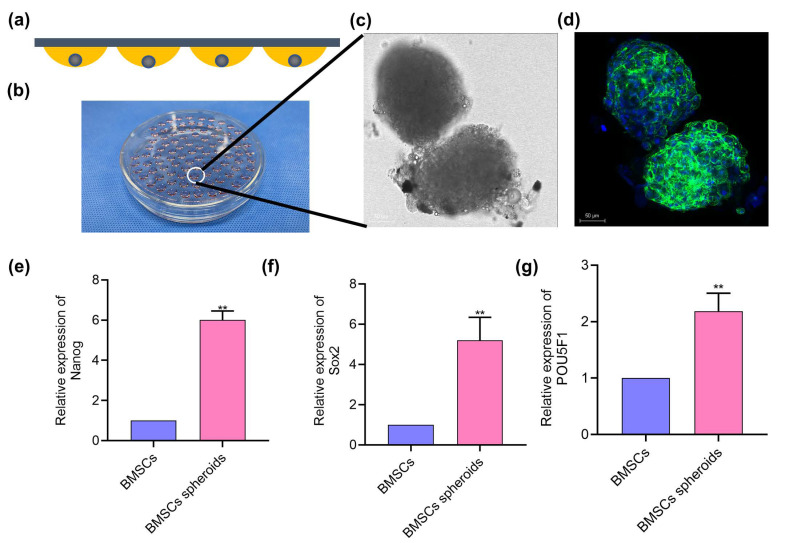
The morphology and stemness-related properties of BMSCs spheroids. (**a**) Schemes of hanging drop method to culture BMSCs spheroids. (**b**) Gross view of the fabrication process of BMSCs spheroids. (**c**) Representative image of BMSCs spheroids. Scale bar = 50 μm. (**d**) Immunofluorescence image of BMSCs using phalloidin and DAPI staining. Scale bar = 50 μm. (**e**–**g**) mRNA expression levels of Nanog, Sox2, and POU5F1 in different cultures of BMSCs. ** *p* < 0.01, vs. BMSCs group (Student’s *t*-test).

**Figure 5 polymers-13-03957-f005:**
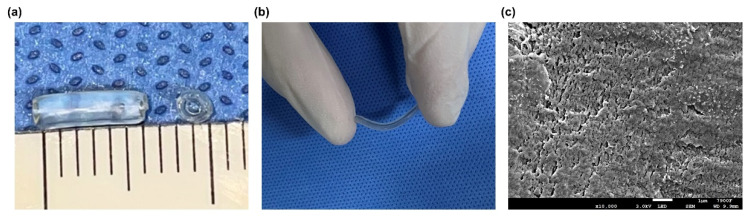
(**a**) Gross view of the chitin nerve conduit. (**b**) The bending image of nerve conduit. (**c**) The scanning electron microscopy image of surface morphology of chitin conduit.

**Figure 6 polymers-13-03957-f006:**
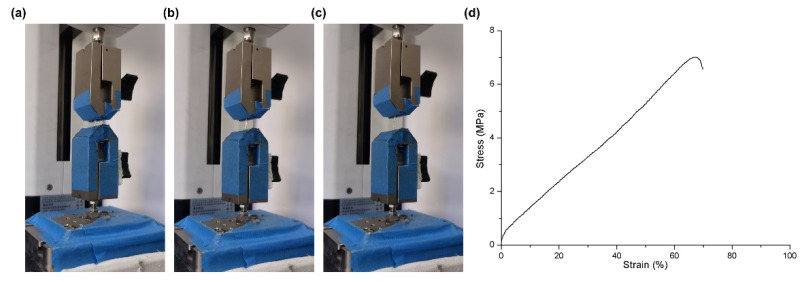
Tensile test of the chitin nerve conduit. (**a**–**c**) The conduit was performing tensile test. (**d**) Stress–strain curve of the chitin nerve conduit.

**Figure 7 polymers-13-03957-f007:**
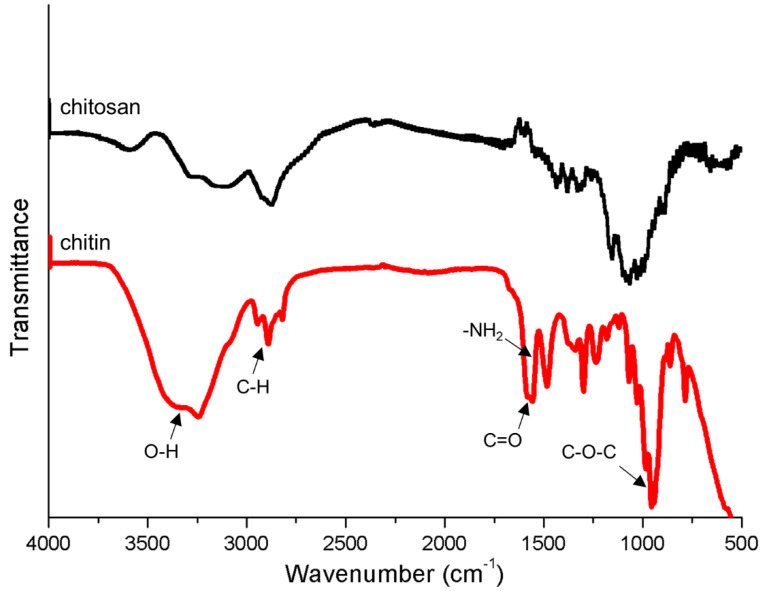
FTIR spectra of chitin and chitosan films.

**Figure 8 polymers-13-03957-f008:**
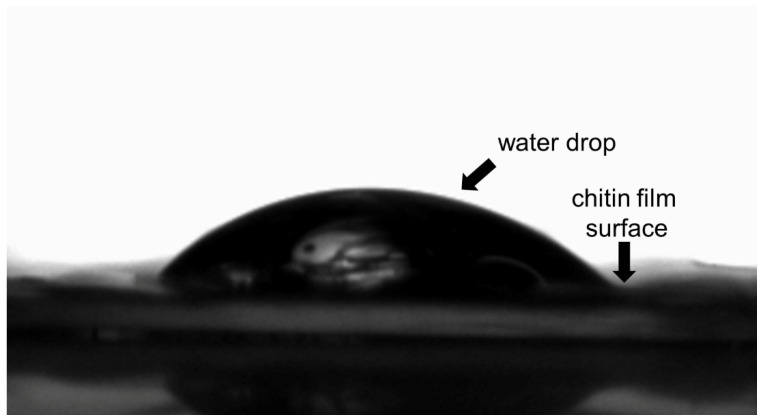
Photograph of the water drop in contact with the surface of chitin film.

**Figure 9 polymers-13-03957-f009:**
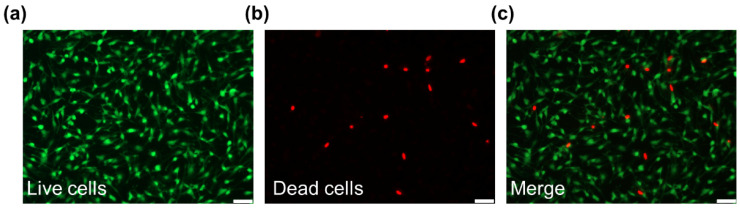
Live/dead staining images of SCs seeded on chitin film. (**a**) The live SCs were labeled with Calcein-AM (green). (**b**) The dead SCs were labeled with PI (red). (**c**) Merged image of live/dead staining. Scale bar = 50 μm.

**Figure 10 polymers-13-03957-f010:**
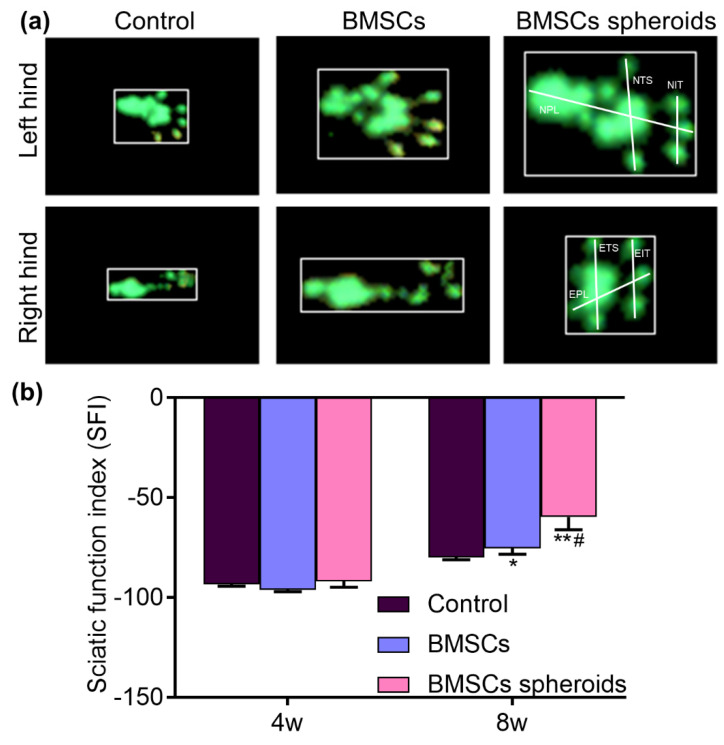
Motor function 4 weeks and 8 weeks after surgery. (**a**) Representative footprints images of right (injured) hind and left (normal) hind paw at 8 weeks after operation. (**b**) The sciatic function index (SFI) recorded at 4 weeks and 8 weeks after operation. Data are expressed as median with interquartile range (*n* = 5 for each group). * *p* < 0.05, ** *p* < 0.01, vs. Control group; # *p* < 0.05, vs. BMSCs group (non-parametric test followed by Kolmogorov–Smirnov and Shapiro–Wilk tests).

**Figure 11 polymers-13-03957-f011:**
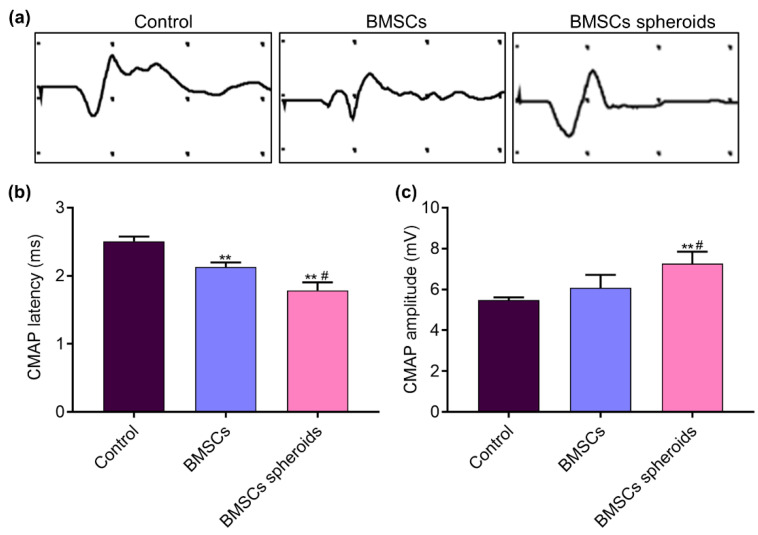
Electrophysiological examinations conducted 8 weeks after surgery. (**a**) Representative CMAP waveform at the operation side in each group. (**b**) Statistical analysis of CMAP latency. (**c**) Statistical analysis of CMAP amplitude. Data are expressed as mean ± SEM (*n* = 5 for each group). ** *p* < 0.01, vs. Control group; # *p* < 0.05, vs. BMSCs group (One-way analysis of variance followed by Kolmogorov–Smirnov and Shapiro–Wilk tests).

**Figure 12 polymers-13-03957-f012:**
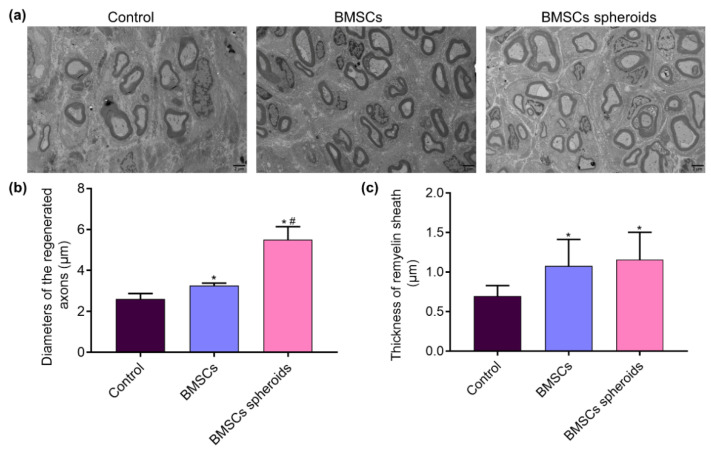
Histological evaluation of the regenerated nerve fibers. (**a**) TEM images of the regenerated sciatic nerve transverse sections 8 weeks postoperatively. (**b**) Diameters of the regenerated nerve fibers. Data are expressed as median with interquartile range (*n* = 5 for each group). (**c**) Thickness of remyelination sheath. Data are expressed as mean ± SEM (*n* = 5 for each group). * *p* < 0.05, vs. Control group; # *p* < 0.05, vs. BMSCs group. Diameters of the regenerated nerve fibers (non-parametric test followed by Kolmogorov–Smirnov and Shapiro–Wilk tests). Thickness of remyelination sheath. One-way analysis of variance followed by Kolmogorov–Smirnov and Shapiro–Wilk tests).

## Data Availability

The data presented in this study are available on request from the corresponding author.

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
