# Peer review of "Chitin Nerve Conduits with Three-Dimensional Spheroids of Mesenchymal Stem Cells from SD Rats Promote Peripheral Nerve Regeneration"

_polymers, 2021, doi:10.3390/polym13223957_

Round 1
Reviewer 1 Report
In this manuscript (polymers-1423461), authors have prepared chitin nerve conduits with three dimensional spheroids of MSCs for promoting peripheral nerve regeneration. This study can be is interesting, but needs major revision for the consideration.
- Please discuss the novelty of this developed nerve conduit for peripheral nerve tissue regeneration, because there are already some published articles on this study. For example, Advanced Engineering Materials, 11(11), 2009, B209-B218. If authors succeed to prove the novelty, then this manuscript can be accepted. Because, reviewer does not see any novelty in this study specifically.
- Introduction section is very poor and needs a major addition of recent published studies based on nerve conduit scaffolds or hydrogels including chitosan and other biomaterials.
- Authors should provide a schematic representation figure for showing effective fabrication of chitin nerve conduits.
- Authors have used acetylation of chitosan scaffolding system, so structural characterization of chitosan and chitin structures should be performed and compared using 1H-NMR, FTIR,and XRD analyses.
- Characterization details should be in detail so that readers may follow the used protocol easily. For SEM analysis, researcher use freeze-drying process to show effective porous-network. Why authors did not use this step in this study?
- Please provide a digital image of tensile testing of chitin conduit for clear understanding in Fig. 3.
- Also, please check whole manuscript for spelling mistakes and grammatical issues. For example, in line 241, TEM should be replaced with SEM.
Reviewer 2 Report
Paper titled (Chitin nerve conduits with three-dimensional spheroids of mesenchymal stem cells promote peripheral nerve regeneration) by Li et al. demonstrated the role of spheroids of mesenchymal stem cells in promoting nerve repair. I think this paper needs major revisions.
1- Title: please mention the animal model used
2- Groups mentions (the hollow chitin conduits, BMSCs and BMSCs-spheroids groups) ; I wish to ask whether there a combined group containing the chitin conduits + BMSC spheroids?
Or there is a missed expression for the groups?
I expect that each group received the conduits but emply or BMSCs or BMSCs spheroids. hence we can consider appropriate controols are achieved.
3-Did authors check the normality of distribution of the data by a suitable test before applying ANOVA test??
4- All methods need adding references
5- Explain how calculated sciatic function index in the METHODS & figure legend
6- Results: Fig 5 and all figures : write in the figure legends what is the type of the presented data (mean, median, ...etc).
7-Please revise English misuse: an example(Line 290: was significance higher than the BMSCs group)
8- what was the age of the rats at the begin of the experiment
9 -Use appropriate abbreviations, h for hours...etc
10- The group names in last figure is different from in methods: please revise all of them and keep a consistent mode all over the manuscript.
Round 2
Reviewer 1 Report
The authors have not responded well to some comments.
- Fabrication process is not well described. Fro example, from line 148 to line 153. Also, the description and schematic figure do not match. In addition, what was the concentration of chitosan used in this study? There are so many flaws in this manuscript. Please check carefully and correct them effectively. Re-draw this schematic figure and provide effective representation.
- Structural characterization is weak. Please characterize this only synthesized product effectively.
- Authors have now added the freeze-drying method to the manuscript. If they used freeze-drying, then where are the porosity and the pore size distribution? Also, Fig. 4(b) is an SEM image, but the authors have written TEM image in the figure caption. Please correct like this mistakes throughout the manuscript. Moreover, Fig. 4 is not observable. Authors should prepare separate figures and incorporate them in the manuscript for clear understanding and observation.
- All figures need good resolution for clear observation.
- In comment 6, Reviewer asked for the photograph of the sample (conduit) while undergoing tensile stretch or strain. Reviewer is very surprised that how this is difficult to perform?. Please provide the digital images at 0 (zero) stage, mid-stage, and fracture-stage tensile strains.
- Also provide a bending image of this conduit for showing its stretchability behavior.
Reviewer 2 Report
Thanks for authors for revising the manuscript. I have a question which was not comprehended in the first revision:
1-Did authors check the normality of distribution of the data by a suitable test like Shapiro Wilk test or K-S test to ensure normal distribution before applying ANOVA test??
2- I wish this complete description for groups [hollow chitin conduits, chitin conduit-integrated BMSCs suspension (BMSCs group) and chitin conduit-integrated BMSCs spheroids group (BMSCs spheroids group)]. appear in figures the same way to be self explanatory. the current view does not show the full idea.
Keep consistent for all manuscript & figures
3- Explain how calculated sciatic function index in the METHODS & figure legend: please specify page and line; this reviewer cannot find it.
Round 3
Reviewer 1 Report
In my opinion, this manuscript can now be accepted for publication.
Reviewer 2 Report
Thanks for the revision
Data which did not conform normal distribution should be expressed as median and quartiles & analyzed by non-parametric ANOVA; revise figures and stats for these data
Mention in the methods that you tested data dist by Shapiro Wilk & KS tests.
